



# Robust handling of extremes in quantile mapping - "Murder your darlings"

Peter Berg[1], Thomas Bosshard[1], Denica Bozhinova[1], Lars Bärring[1], Joakim Löw[1], Carolina Nilsson[1], Gustav Strandberg[1], Johan Södling[1], Johan Thuresson[1], Renate Wilcke[1], and Wei Yang[1]

[1]Swedish Meteorologcal and Hydrological Institute, Folkborgsvägen 17, 601 76 Norrköping, Sweden

**Correspondence:** Peter Berg (peter.berg@smhi.se)

**Abstract.** Quantile mapping is a method often used for bias adjustment of climate model data toward a reference, i.e. to construct a transformation of the model's distribution to that of the reference. The main moments of the distributions are typically well transformed by quantile mapping, but statistical uncertainty increases towards the extreme tails, making robust transformations challenging. Because of the limited data at the extreme tails, also an empirical quantile mapping needs to make some estimation or fit a parameterized function for data beyond the calibration data range. The MIdAS bias adjustment platform is here employed to explore different methods for handling the extreme tail, which are evaluated using an indicator for extreme precipitation - the maximum daily precipitation amount per year. Different methodologies are evaluated for a large ensemble of regional climate model projections over Scandinavia. The sensitivity of the empirical quantile mapping for the tails of the distribution is demonstrated, and it is found that the behaviour is significantly different within and without the calibration period, causing severe issues with the temporal consistency of the timeseries. The sensitivity is identified to be due to differences in the activated features of the bias adjustment, within the calibration period where the empirical transfer function is applied, and outside that period, where the extrapolation method is likely applied. This means that the bias adjustment method is in a sense different between different time periods. Currently MIdAS uses separate calibrations for each day of the year, as opposed to e.g. for each calendar month, further aggravates this issue. Further, finding a robust parametrisation for the tail is not straightforward. We identify a two-step solution which works well for this problem: (i) "murder your darlings" by excluding data from the tail data in the calibration period, the extrapolation feature is activated for all time periods, even the calibration period, and (ii) applying an outlier insensitive method for linear regression works well for finding an extrapolation parametrisation for the tail.

## 1 Introduction

Bias adjusted climate projections are routinely used for impact modelling, and further processed into climate indicators for various climate services. Climate indicators of extremes are, by definition, sensitive to the small sample, which becomes even more sensitive when combined with a reference data to map a transformation in the bias adjustment step. Such sensitivity can impose large uncertainties in the interpretation and conclusions drawn from the extreme indicator, in worst case rendering the information useless or even misleading.





Earlier studies have identified issues with bias adjusting data outside the calibration range for pure empirical quantile mapping approaches (Boé et al., 2007; Bellprat et al., 2013). A common solution is to apply the adjustment value of the high end of the calibration period for all data outside the calibration range (Themeßl et al., 2011), although this may introduce unrealistically large adjustments (Switanek et al., 2017). A combination of empirical and parametric approaches have been proposed by several authors, e.g. by using extreme value theory fits to the top 5% data (Tani and Gobiet, 2021), while others have applied
linear fits to the extremes (Holthuijzen et al., 2022).

Clearly, a purely empirical method based on all data in the calibration period will act differently when applied outside the calibration data range compared to its calibration. Assumptions on how the method should react to new data, e.g. re-using the highest adjustment value of the calibration range (Themeßl et al., 2011), will not be the same method as that which was calibrated. This may lead to unexpected results for the bias adjusted tails. One can force the bias adjustment to apply its
full range of methods only by making sacrifices at the very tail of the distribution. In a way similar to the literary method of "Murder your darlings" (Quiller-Couch, 2015), also known as "kill your darlings", i.e. to remove the most precious items for the greater good of the work: "Whenever you feel an impulse to perpetrate a piece of exceptionally fine writing, obey it—whole-heartedly—and delete it before sending your manuscript to press. Murder your darlings."

This paper presents a clear example of problematic side effects of bias adjustment within and outside the calibration period.
A new method to handle the calibration strategy and distribution fits to the tail is presented and tuned to find a pragmatic use of data while reducing the side-effects. The example is based on data from the Swedish climate service, using a large ensemble of regional climate models and the MIdAS bias adjustment method (Berg et al., 2022).

## 2    Bias adjustment

The MIdAS implementation of quantile mapping starts from the quantile-quantile (Q-Q) plots of the reference and model data
sets, which share the same number of data points. A piece-wise linear smoothing spline function is fitted to the Q-Q plot, see Berg et al. (2022) for details. MIdAS applies a linear function fitted to the 90% most central data points of the Q-Q plot, with weights defined by the standard deviation of the data points from the linear fit. A linear continuation of the spline is applied to data points outside of the calibration data range, i.e. a "$1 - 1$" linear continuation of the spline in the Q-Q plot.

The transfer functions are calculated based on a historical period, here 1971–2000, for each grid point and in sub-sets of the
annual cycle. Rather than using calendar month sub-sets, as in most published methods, MIdAS is set up to calculate and apply the transfer functions based on the day of the year ($doy = [1, 365]$), using a moving window of 15 $d$ before and after $doy$, such that 31 $d$ are used to build the distribution of the reference and model data.

### 2.1    New parameterization for the tail

The new development to handle data at the tails of the distributions is based on the fitting procedure of Theil-Sen (Theil,
1950; Sen, 1968) which is an outlier insensitive method. The procedure is to calculate the median of slopes derived from each





individual pair of points in the sample, i.e. in the Q-Q plot. It means that outliers will have little individual effect on the fits, making the linear fits robust to the high sample uncertainties that are unavoidable at the tail of the distributions.

When excluding high extreme data points in the calibration sample of precipitation, in order to activate extrapolation behaviour and the full bias adjustment method, there are unavoidable effects on other moments of the distribution. Because
precipitation extremes often add significant quantities of precipitation, they are important in defining the mean moment. A balance between a good handling of extremes, and a good adjustment of the mean moment must be found.

Different versions of excluding data from the calibration data range are combined with the Theil-Sen regression to find a balance between side-effects on the tail data and mainly the mean moment of the bias adjusted data:

- $R0T5$ - no excluded data and calibration on percentiles 95–100

– $R1T5$ - exclude 1% of the data on the upper tail and calibrate on percentiles 94–99

- $R5T5$ - exclude 5% on the upper end and calibrated on percentiles 90–95.

## 2.2 Data

Precipitation data from SMHIGridClim (Andersson et al., 2021) is used as reference data for the bias adjustment. SMHI-GridClim is a data set based on the regional reanalysis UERRA (UERRA, 2019) combined with gauge data from Sweden
and neighbouring countries, mapped at a 2.5 km grid and with daily temporal resolution. For this analysis, the data set is conservatively remapped to the Euro-CORDEX 0.11 degree (approximately 12.5 km) grid covering Scandinavia.

The climate projections are acquired from the Euro-CORDEX CMIP5 data set (Jacob et al., 2020). A large ensemble of 67 unique combination of global and regional models are used (see Table 1), using the RCP8.5 scenario of future emissions. The ensemble members all have bias to different extent both for the mean and the extreme tails, as evaluated for a sub-set of the
ensemble in, e.g., Vautard et al. (2021).

## 2.3 Evaluation methods

Two statistics are used to evaluate the different methods in Sec. 2.1: the annual mean, and the annual maximum of daily precipitation. The mean is evaluated because it is summarizing the performance of the bias adjustment across all data, while the annual maximum highlights the most extreme values, that are specifically targeted in this study. Because the signal-to-
noise levels are very high for the annual maxima, the ensemble mean is calculated across all members, and in addition a spatial average is calculated over the land regions of the complete domain. The main figures present the temporal evolution of ensemble mean for the the domain average annual mean and maxima.

## 3 Results

Figure 1 shows the original ensemble result of annual maxima of daily precipitation averaged over the domain, together with
the reference data and the resulting bias adjusted data using the original implementation of MIdAS, as presented in Berg et al.

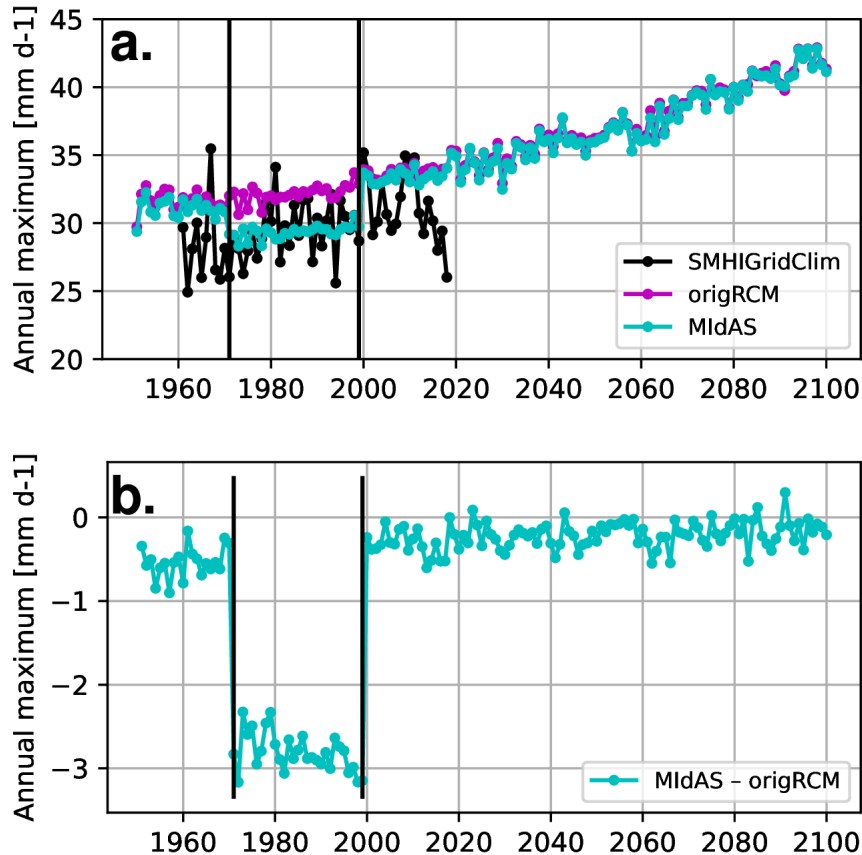

**Figure 1.** Annual precipitation maxima (mm d$^{-1}$) for SMHIGridClim, the original RCM ensemble mean, and the bias adjusted data using the standard MIdAS setup: for absolute values (a), and for the difference between bias adjusted and original model data (b).

(2022). In the calibration period, marked with vertical bars, the bias adjustment is efficiently offsetting the annual maxima to be at a similar level as the reference. Note that the interannual variability is reduced due to the ensemble mean for the model data. However, outside the calibration period, there is almost no visible effect of the bias adjustment, resulting in significant discontinuities at the beginning and the end of the calibration period. This is clearly an issue, and is, as will be shown, caused by the essentially different bias adjustment methods within (without extrapolation) and outside (with extrapolation) of the calibration period. The issue is very clearly seen in Fig. 1 because of the averaging over a larger domain. When assessed for single grid points or smaller domains, the issues are hidden within the large noise levels for this kind of extreme precipitation statistics. This highlights the need to quality control and evaluate bias adjustment across larger areas, even though the parametrization and scale of the bias adjustment is intended for single grid points. Clearly, if the calibration period would also be used as a historical reference period, the climate change signal would be exaggerated by almost 3 $mmd^{-1}$, which is about twice the signal from the original data at mid-century.



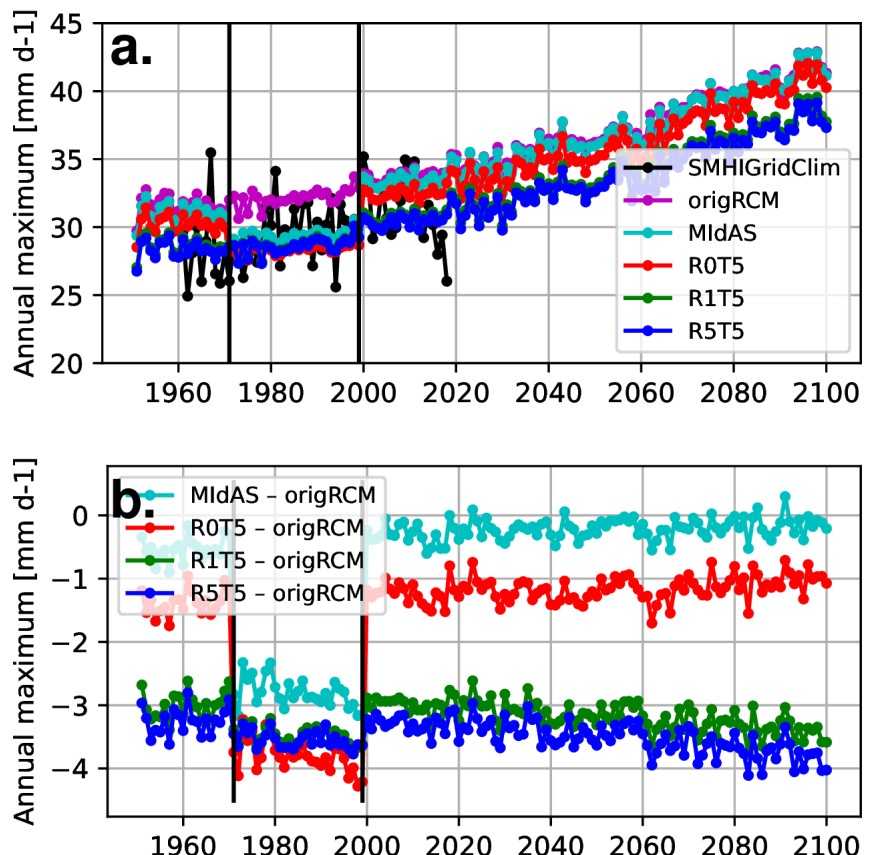

**Figure 2.** Same as Fig. 1, but with the additional data sets for the experiments R0T5, R1T5 and R5T5. Note that the green lines lies behind the blue line in panel a.

In an attempt to improve on the performance outside the calibration period, the Theil-Sen method is applied to find a good fit to the top 5% data of the distribution (experiment R0T5), which is shown as a red line in Fig. 2. The adjustment within the calibration period is only mildly affected, due to the change from an assumed linear extrapolation in the original MIdAS method, and the Theil-Sen methodology. However, the main issue remains as there is still a significant offset at the beginning and the end of the calibration period.

Because the bias adjustment method will inevitably activate the extrapolation routine with data outside the calibration range, the next experiments (R1T5 and R5T5) forces the extrapolation to be active also within the calibration range. In other words, some extremes are excluded for the benefit of an overall better adjustment, at the likely cost of worse performance in the calibration period. Combining the Theil-Sen fit with exclusion of the top 5% of the calibration data (R5T5, blue) has a strong impact on the bias across the whole time series. The bias is still well adjusted in the calibration period, equal to the R0T5 experiment, but with the additional much improved performance outside the period. This result indicates that only by "murdering

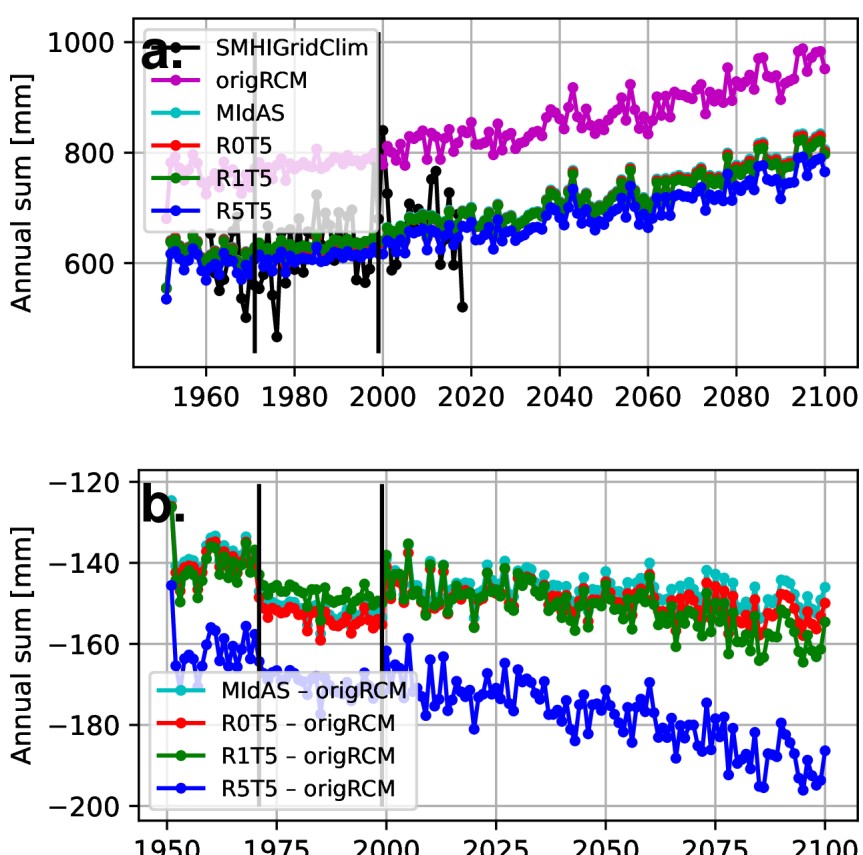

**Figure 3.** Same as Fig. 2, but for the annual sum of precipitation. Note that R0T5 is very close to R1T5 in panel a.

your darlings", one can reach a consistent bias adjustment across the time periods, i.e. by activating the extrapolation routine for all periods. Similar results are seen for the experiment R1T5 (green), where less data (1%) is excluded.

So what are the side-effects? The annual maxima is but one of many important aspects that the bias adjustment is supposed to improve, and often the more accumulated statistics such as the mean values are more important to reproduce. Figure 3 shows the annual sums of precipitation, i.e. the result on the accumulated precipitation of all intensities. While the original MIdAS method works well for this measure, the R5T5 method clearly imposes a dry bias. This is because a significant amount of precipitation has been removed from the distribution in the calibration period (the 5% highest intensity events), which strongly

impacts the overall bias. The sign of this impact depends on the original bias in the mean and the maximum precipitation. They are likely of the same sign as the maximum strongly affects the mean, but it may not always be that way. Reducing the exclusion of data to 1% (R1T5), the annual sums are closer to the reference data set, and the original method (Fig. 3, and as presented above, still resulting in a much similar result for the adjustment of the annual maxima (Fig. 2). However, Fig. 3 also highlights another important side-effect - a reduced trend of the increasing precipitation with time; most clearly seen in the





difference plot Fig. 3b. This trend is seen for the original MIdAS, as well as all the experiments, although the impact seems significantly stronger for experiment R5T5. No significant impact on trends in annual maxima is seen for the original MIdAS adjustment and experiment R0T5, see Fig. 2b. However, when data are excluded in experiment R1T5 and R5T5, a negative impact is introduced; much similar to that imposed on the mean statistic. One can debate whether this is a side-effect, or whether it is good to have consistent behaviour across the statistics, i.e. that the relative effects of the extremes and the means

more closely follow each other in time.

## 4   Conclusions

This study focuses on an identified issue with bias adjustment of the highest extremes, which are adjusted differently within and outside the calibration period. A new outlier insensitive linear fit is used for the extreme tails, and a solution to the issue is presented in a set of experiments. The main conclusions are:

– The more extreme the statistic of interest is the more elusive any bias becomes as the available data becomes scarcer and bias is a fundamentally statistical property. To create a robust sample size the bias should be assessed over an ensemble of simulations and/or over a larger set of gridcells.

– A consistent bias adjustment method must have all features activated across all time periods, including the calibration period, in order to produce consistent bias adjustment.

– The extrapolation feature can be activated by excluding the highest data points in the calibration period, making sure that the extrapolation feature is acting on the complete time range, and result in consistent bias adjustment.

– An unavoidable trade-off between adjustment of the mean moment and the extremes is necessary, as excluding high intensity data points from the calibration will inevitably affect the mean.

– As there is an ever increasing focus on climate extremes we suggest that the performance of bias adjustment methods
should routinely include an assessment of its impact on the extreme tails

*Code and data availability.* The MIdAS git repository is open for all to access and use under the GNU LESSER GENERAL PUBLIC LICENSE v3, at https://git.smhi.se/midas/midas. The code used for the final setup that handles the extreme tails is implemented in v0.3.0. The annual maxima and mean values, as well as Python scripts for reproducing the figures are available in Berg and Södling (2024).

*Author contributions.* PB – Conceptualization, Methodology, Formal analysis, Writing – Original Draft, Project administration, Funding
acquisition; TB – Methodology, Software, (Writing – Review and Editing); LB – Methodology, (Writing – Review and Editing); JS – Methodology, Formal analysis, Investigation, Software, (Writing – Review and Editing); RW - Methodology, (Writing – Review and Editing); DB – Software, (Writing – Review and Editing); JL – Software, (Writing – Review and Editing); CN – Software, (Writing – Review and





Editing); JS – Software, (Writing – Review and Editing); JT – Software, (Writing – Review and Editing); WY – Software, (Writing – Review and Editing); GS – Conceptualization, (Writing – Review and Editing), Project administration, Funding acquisition.

*Competing interests.*  The authors declare that they have no conflicts of interest.

*Acknowledgements.*  We acknowledge funding from the Swedish Meteorological and Hydrological Institute. Analyses were performed on the Swedish climate computing resource Bi provided by the Swedish National Infrastructure for Computing (SNIC) at the Swedish National Supercomputing Centre (NSC) at Linköping University. We acknowledge the work by Klaus Zimmermann in writing the base code, as presented in Berg et al. (2022), setting up the git repository, and much more.





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



**Table 1.** List of the EURO-CORDEX GCM-RCM simulations included in the evaluation and the RIP (realisation-initialization-physics) code.

| GCM | RCM | RIP | GCM | RCM | RIP |
|---|---|---|---|---|---|
| CCCma-CanESM2 | CCLM4-8-17 | r1i1p1 | MOHC-HadGEM2-ES | ALADIN63 | r1i1p1 |
| CCCma-CanESM2 | REMO2015 | r1i1p1 | MOHC-HadGEM2-ES | HIRHAM5 | r1i1p1 |
| CNRM-CERFACS-CNRM-CM5 | COSMO-crCLIM-v1-1 | r1i1p1 | MOHC-HadGEM2-ES | REMO2015 | r1i1p1 |
| CNRM-CERFACS-CNRM-CM5 | ALADIN63 | r1i1p1 | MOHC-HadGEM2-ES | RegCM4-6 | r1i1p1 |
| CNRM-CERFACS-CNRM-CM5 | HIRHAM5 | r1i1p1 | MOHC-HadGEM2-ES | WRF381P | r1i1p1 |
| CNRM-CERFACS-CNRM-CM5 | REMO2015 | r1i1p1 | MOHC-HadGEM2-ES | RACMO22E | r1i1p1 |
| CNRM-CERFACS-CNRM-CM5 | WRF381P | r1i1p1 | MOHC-HadGEM2-ES | HadREM3-GA7-05 | r1i1p1 |
| CNRM-CERFACS-CNRM-CM5 | RACMO22E | r1i1p1 | MOHC-HadGEM2-ES | RCA4 | r1i1p1 |
| ICHEC-EC-EARTH | COSMO-crCLIM-v1-1 | r12i1p1 | MPI-M-MPI-ESM-LR | COSMO-crCLIM-v1-1 | r1i1p1 |
| ICHEC-EC-EARTH | COSMO-crCLIM-v1-1 | r1i1p1 | MPI-M-MPI-ESM-LR | COSMO-crCLIM-v1-1 | r2i1p1 |
| ICHEC-EC-EARTH | COSMO-crCLIM-v1-1 | r3i1p1 | MPI-M-MPI-ESM-LR | COSMO-crCLIM-v1-1 | r3i1p1 |
| ICHEC-EC-EARTH | CCLM4-8-17 | r12i1p1 | MPI-M-MPI-ESM-LR | CCLM4-8-17 | r1i1p1 |
| ICHEC-EC-EARTH | HIRHAM5 | r12i1p1 | MPI-M-MPI-ESM-LR | ALADIN63 | r1i1p1 |
| ICHEC-EC-EARTH | HIRHAM5 | r1i1p1 | MPI-M-MPI-ESM-LR | HIRHAM5 | r1i1p1 |
| ICHEC-EC-EARTH | HIRHAM5 | r3i1p1 | MPI-M-MPI-ESM-LR | REMO2015 | r3i1p1 |
| ICHEC-EC-EARTH | REMO2015 | r12i1p1 | MPI-M-MPI-ESM-LR | RegCM4-6 | r1i1p1 |
| ICHEC-EC-EARTH | RegCM4-6 | r12i1p1 | MPI-M-MPI-ESM-LR | WRF381P | r1i1p1 |
| ICHEC-EC-EARTH | WRF381P | r12i1p1 | MPI-M-MPI-ESM-LR | RACMO22E | r1i1p1 |
| ICHEC-EC-EARTH | RACMO22E | r12i1p1 | MPI-M-MPI-ESM-LR | HadREM3-GA7-05 | r1i1p1 |
| ICHEC-EC-EARTH | RACMO22E | r1i1p1 | MPI-M-MPI-ESM-LR | REMO2009 | r1i1p1 |
| ICHEC-EC-EARTH | RACMO22E | r3i1p1 | MPI-M-MPI-ESM-LR | REMO2009 | r2i1p1 |
| ICHEC-EC-EARTH | HadREM3-GA7-05 | r12i1p1 | MPI-M-MPI-ESM-LR | RCA4 | r1i1p1 |
| ICHEC-EC-EARTH | RCA4 | r12i1p1 | MPI-M-MPI-ESM-LR | RCA4 | r2i1p1 |
| ICHEC-EC-EARTH | RCA4 | r1i1p1 | MPI-M-MPI-ESM-LR | RCA4 | r3i1p1 |
| ICHEC-EC-EARTH | RCA4 | r3i1p1 | NCC-NorESM1-M | COSMO-crCLIM-v1-1 | r1i1p1 |
| IPSL-IPSL-CM5A-MR | HIRHAM5 | r1i1p1 | NCC-NorESM1-M | ALADIN63 | r1i1p1 |
| IPSL-IPSL-CM5A-MR | REMO2015 | r1i1p1 | NCC-NorESM1-M | HIRHAM5 | r1i1p1 |
| IPSL-IPSL-CM5A-MR | WRF381P | r1i1p1 | NCC-NorESM1-M | REMO2015 | r1i1p1 |
| IPSL-IPSL-CM5A-MR | RACMO22E | r1i1p1 | NCC-NorESM1-M | RegCM4-6 | r1i1p1 |
| IPSL-IPSL-CM5A-MR | RCA4 | r1i1p1 | NCC-NorESM1-M | WRF381P | r1i1p1 |
| MIROC-MIROC5 | CCLM4-8-17 | r1i1p1 | NCC-NorESM1-M | RACMO22E | r1i1p1 |
| MIROC-MIROC5 | REMO2015 | r1i1p1 | NCC-NorESM1-M | HadREM3-GA7-05 | r1i1p1 |
| MOHC-HadGEM2-ES | COSMO-crCLIM-v1-1 | r1i1p1 | NCC-NorESM1-M | RCA4 | r1i1p1 |
| MOHC-HadGEM2-ES | CCLM4-8-17 | r1i1p1 | | | |