# Peer review of "Robust handling of extremes in quantile mapping - "Murder your darlings""

_Geoscientific Model Development, 2024_

## Referee Comment (RC1)

gmd-2024-98 manuscript 'Robust handling of extremes in quantile mapping - "Murder your darlings"' by Berg et al.

GMD - Review criteria (geoscientific-model-development.net) :

1. Does the paper address relevant scientific modelling questions within the scope of GMD? Does the paper present a model, advances in modelling science, or a modelling protocol that is suitable for addressing relevant scientific questions within the scope of EGU?

Yes, it suggested and important fix to the MIDAS method, and the comments made are relevant for a wide range of bias-correction methods that deal with extremes and the problem of extrapolating these.

2. Does the paper present novel concepts, ideas, tools, or data?

Yes.

3. Does the paper represent a sufficiently substantial advance in modelling science?

Yes - important for climate adaptation efforts worldwide.

4. Are the methods and assumptions valid and clearly outlined?

Yes. The paper is directed at people who know what bias corrections are and who appreciate the special problem at the extremes, but is spot on for this audience.

5. Are the results sufficient to support the interpretations and conclusions?

Yes.

6. Is the description sufficiently complete and precise to allow their reproduction by fellow scientists (traceability of results)? In the case of model description papers, it should in theory be possible for an independent scientist to construct a model that, while not necessarily numerically identical, will produce scientifically equivalent results. Model development papers should be similarly reproducible. For MIP and benchmarking papers, it should be possible for the protocol to be precisely reproduced for an independent model. Descriptions of numerical advances should be precisely reproducible.

Yes. The updated MIADAS code is available online.

7. Do the authors give proper credit to related work and clearly indicate their own new/original contribution?

Yes.

8. Does the title clearly reflect the contents of the paper? The model name and number should be included in papers that deal with only one model.

Well, maybe stick 'MIdAS' in there somewhere, but that could mar the nice title!

9. Does the abstract provide a concise and complete summary?

Yes.

10. Is the overall presentation well structured and clear?

Yes.

11. Is the language fluent and precise?

Yes - a few typos etc noted in this review.

12. Are mathematical formulae, symbols, abbreviations, and units correctly defined and used?

NA.

13. Should any parts of the paper (text, formulae, figures, tables) be clarified, reduced, combined, or eliminated?

No. Sweet and short is fine.

14. Are the number and quality of references appropriate?

Yes.

15. Is the amount and quality of supplementary material appropriate? For model description papers, authors are strongly encouraged to submit supplementary material containing the model code and a user manual. For development, technical, and benchmarking papers, the submission of code to perform calculations described in the text is strongly encouraged.

The updated code is available online, so that's fine.

---

## Author Response (AR1)

**CEC1**: 'Comment on gmd-2024-98', Juan Antonio Añel, 20 Jun 2024

Dear authors,

Unfortunately, after checking your manuscript, it has come to our attention that it does not comply with our "Code and Data Policy".

https://www.geoscientific-model-development.net/policies/code_and_data_policy.html

You have archived your code on a git repository which is on a server not suitable for scientific publication. Also, you have to publish and share openly all the data used in your study, not only the data to recreate the plots. This includes the SMHIGridClim and Euro-CORDEX CMIP5 data used. We understand that some files used in your study are large (e.g., full output from models). In such cases, instead of storing the complete files, you should at least keep the variables or final fields computed and used in your work.

Also, you must include in a potentially reviewed version of your manuscript the modified 'Code and Data Availability' section, DOIs and links for the new code and data repositories.

Please, reply as soon as possible to this comment with the link and DOIs for it so that it is available for the peer-review process, as it should be.

Please, be aware that failing to comply promptly with this request could result in rejecting your manuscript for publication.

Juan A. Añel

**Dear Juan A. Añel,
thanks for the clarifications of the requirements of the journal, and please accept our sincere apologies for not fulfilling them in the first submission of the manuscript. We updated the repository of the paper, and where we formerly only had the information necessary to reproduce the final analysis and figures of the paper, we have complemented with all the raw timeseries needed to reproduce the experiments, a zip-file containing the published MIdAS souce code, as well as instructions for how to perform the experiments. The reference to the data repository is updated in the revised manuscript.**

**RC1**: 'Comment on gmd-2024-98', Anonymous Referee #1, 24 Jun 2024
Review of gmd-2024-98 manuscript 'Robust handling of extremes in quantile mapping - "Murder your darlings"' by Berg et al.

General comments:

The paper gives an interesting insight into the special problems of extreme-value extrapolations/bias corrections. The paper identifies a real problem in the MIdAS method which is also important for efforts not based on that specific software. A series of tests were performed and new approaches are suggested.

Important paper that should be published just to help users of MIdAS. The ideas in the paper will also provide food for thought for general practitioners in the field of climate-change projection; especially at the extremes end of the distributions.

**Thank you very much for the positive and detailed review. We answer below to each of the issues raised. We would like to clarify that the presented issue is not specific for the MIdAS implementation of quantile mapping, but is inherent to all empirical quantile mapping methods and is a fundamental issue that needs to be dealt with. It is, however, not easily discovered due to the large noise levels, which is why we used a large ensemble of models and averages over a larger domain to visualize the issue.**

Specific comments:

The paper is set in a language attuned to people who use MIdAS, which is fine for that community, as well as accessible for people in the relevant areas.

I'd like to see a sentence or two more on the subject in Line 71 'conservatively remapped' - how?

**We specified the methods by adding "(using first order conservative remapping following \citet{Jones1998})"**

Technical corrections:

Line 22: 'a reference data' -> perhaps just 'reference data'
**Done.**

Line 23: 'in worst case' -> 'in the worst case'
**Done.**

Line 31: 'will act differently' - 'will'? - perhaps true. Seems a strong statement.
**Changed to "might".**

Line 81: 'main figures' -> 'main Figures'
**We removed "main", but interpret the journal guidelines to use capitalized version only when a specific figure is referenced.**

Figure 1 caption: 'absolute values'? Do you mean numerical absolute, or a more general phrase relating to 'levels' (as opposed to anomalies)?

**Changed to "absolute levels" as suggested.**

Line 87: 'due to the ensemble mean' … elaborate that thought, perhaps?
**We clarified by changing to "the ensemble averaging performed for the model data".**

Figure 2 legend in both panels, but especially the lower panel, obscures the graphs.
**We have adjusted the figure legends to better show the data.**

Line 103: 'forces the' -> 'force the', as 'experiments' is plural.
**Done.**

Line 108 was hard for this reviewer to parse.
**We reformulated to "This result indicates that one can only reach a consistent bias adjustment across the time periods by activating the extrapolation routine for all periods, which implies disregarding of some tail data. In other words, by "murdering your darlings"."**

**RC2**: 'Comment on gmd-2024-98', Anonymous Referee #2, 26 Aug 2024
I liked this paper. It is well written, succinct, has a catchy title, and makes an important point.

**Thank you very much for the positive and detailed review. We answer below to each of the issues raised.**

My one complaint – and it is a big complaint – is that this paper only considers linear extrapolation. My intuition is that you wouldn't need to "murder your darlings" if you fit an extreme value distribution to the tail of the data. I would strongly recommend adding the results from such an implementation to Fig. 2 and Fig 3. I'd be ok accepting the paper without this addition, however, since it already stands on its own as a cautionary tale.

**There is a background philosophy of MIdAS that was not well stressed in the current paper, but more so in the original reference. The philosophy is to have a method that is well applicable across the globe and across seasons. As soon as a distribution is assumed, it becomes problematic to an operational method to follow up on cases where the distribution is not justified. This can occur for different regional climates, or for specific seasons and climate regimes. For this reason, we develop the MIdAS code to be transparent and preferably of low-complexity. The linear model (in the QQ-plot), implicitly assumes that the climate model performs generally in line with the reference data for the tail, although the magnitudes might differ, i.e. a linear slope deviating from the 1:1 line in in the QQ-plot.**

**The main reason for not using a more standard extreme value theory, such as generalized extreme value or peak-over-threshold, is because of the added complexity it implies for the implementation, the transparency of which distribution (and parameter set) that is used, and the goodness-of-fit in different locations and times. We believe that such approaches will introduce added uncertainty in larger production jobs, and the more simple linear method is preferred, although it might in some occasions have worse performance.**

**We added a text-block to emphasize this point in the description of the new parameterization: "The Theil-Sen approach aligns with the general philosophy of MIdAS to use generally applicable methods that are not dependent on specific distributions. The reason for this philosophy is that MIdAS shall be transparent and equally applicable across geographic regions and climates without the need to pre-define specific distribution functions for each case."**

One other big-picture thought is that the efficacy of your extrapolation method at reproducing the real world can be assessed by removing the top 5 percent of your training data and confirming that your extrapolation method correctly predicts that top 5% during the training period. Fortuitously, your Fig 2b shows this in the sense that MIdAS minus origGCM (teal line) by construction handles the top 5% properly during the training period and R5T5 (red line) is what happens when you throw out the top 5% of the data and try to extrapolate it instead. The difference between the teal line and the red line represents the failure of your extrapolation to produce the correct extreme values. It doesn't look to me like your method is doing a very good job; again, I'm curious whether an extreme value distribution fit would perform better.

The reviewer makes a good point, and indeed the new methods with a linear fit "RxT5" do in general perform worse than the original "MIdAS" method for the reference period, as shown in figure 1 below. We have made an analysis on how well the different versions perform for the annual maxima, which is shown in the figure below. In this figure we compare the mean annual maxima for the calibration period with the reference data, and make a box plot for all the ensemble members. Clearly, there is a systematic underestimation in the Theil-Sen methods compared to the original MIdAS method, but not far away from the original empirical method by less than 1 mm d$^{-1}$ (or less than 0.5% relative bias). Thus, we argue that the linear assumption is acceptable, seen also in light of the answer to the first topic above.

[Figure]

Figure 1: The mean annual maxima calculated for the calibration period, and presented as absolute bias (mm d$^{-1}$) from the reference data (SMHIgridclim), as box-plots of all ensemble members.

We added a paragraph at the beginning of the Results section to present this result (using the figure above), and with the following text:
"Figure~\ref{fig:performance_plot} shows the performance of the different MIdAS setups for the annual maxima. The original MIdAS code is close to the reference data, which is expected as all the data points are included, and the deviations for different ensemble members is due to how well the spline is fitted to the tail of the distribution. The different Theil-Sen methods show similar behaviour across the ensemble, but with a general underestimation of the annual maxima after bias adjustment. The remaining bias is on the order of less than 1~mm d$^{-1}$ for the mean of the ensemble, which is less than o.5\% relative bias. We consider this a sufficiently good fit, which does not urge for using more advanced fitting methods using extreme value theory."

Other than these big-picture complaints, I have only minor complaints about wording:

L14: missing word: "for each calendar month, WHICH further aggravates this issue"

Done.

I think it would be useful if you define quantile mapping in the introduction. This will make your paper accessible to people who are just learning about bias correction.

We have added a sentence in the introduction where we point to further details of the definition in our earlier paper on MIdAS: "Many bias adjustment methods are based

on the quantile mapping approach, where a transfer function is used to map model data in different quantiles of a distribution to match that of the reference data set, see further detailed descriptions in \citet{Berg2022}.”

L33-34: “…will not be the same method as that which was calibrated…” Could you rewrite this sentence? I don’t understand it at all.

**We have reformulated this part: “Clearly, a purely empirical method based on all data in the calibration period might act differently when applied outside the calibration data range compared to its calibration. The method reacts to data outside the calibration range differently to inside the range, e.g. re-using the highest adjustment value of the calibration range \citep{Themessl2011}, which means that the behaviour of the bias adjustment differs depending on the magnitude of the values that are adjusted. In other words, the bias adjustment method differs for data inside and outside the calibration range.”**

L35: “One can force the bias adjustment to apply its full range of methods only by…” – I don’t think “methods” is the right word?

**We clarified by changing to “its full behaviour”.**

~L47: I think it would be useful to write out the equation for your linear spline implementation.

**The details of the spline fit is not central to the topic discussed in the current paper, and we have instead added a reference to another paper where we describe the method in detail: “, as explained in detail in \citep{Berg2022}.”**

L52: “31 d are used to build the distribution of the reference and model data” – I think you mean 31 days times 29 years of samples, right?

**Yes, we corrected this: “such that 31~$d$ times the number of calibration years are used”.**

L65: Since you always use Theil-Sen regression for the top 5% of the data, you could remove the “R5” part of your T%R5 names if you want.

**That is correct. However, we decided to keep them as we want to emphasize that we are still using 5% of the data in all experiments, although they are based on different quantile.**

L73: put an “s” on the end of “unique combination”

**Done.**

Figs 1-3: in panel b, it would be useful if the y label could be clear that this is a **difference** in annual maxima.

**We believe that the captions are clear in that we present annual maxima (or mean in Fig.3), and that panels (a) show absolute levels, while (b) shows difference between adjusted and original.**

L87: "interannual variability is reduced due to the ensemble mean for the model data" – it may be worth explicitly clarifying that reduced variance in no way indicates that the quantile mapping is wrong – it is due to averaging realizations once the remapping is done.

**Yes, this was also noted by another reviewer, and we have changed to "due to the ensemble averaging performed for …"**

I'd like to know a bit more about why the original MIdAS approach causes bias corrected future predictions to look ~identical to the original data. It seems like this would only happen if the slopes you're using for extrapolation are ~identical for the model data and the observation data, which doesn't seem assured. Could you say more about this?

**This feature is indeed interesting, and we have no complete explanation of this. We believe that it is a consequence of the reference data being based on a model simulation as baseline before including the station information. Over large areas, the reference data will therefore be "model like" in its description of extremes, which means that the QQ-plots tend to lie on the linear 45 degree slope assumed in the original method. This gives a general offset to the values when the extrapolation is activated outside the calibration period.**

~L63: you could be more clear that you are doing a *single* Thiel-Senn regression using all data from the top 5% of the remaining data.

**We stated this more clearly: "…Theil-Sen regression on the top 5\% data…"**

On L123 you say that changed slope is a "negative impact" but the following sentence says that whether it is good or bad is up for debate. This is inconsistent.

**We reformulated to clarify this: "However, when data are excluded in experiment R1T5 and R5T5, there is also an impact on the trends; much similar to that imposed on the mean statistic."**

L131: The second sentence of this bullet ("To create a robust sample size the bias should be assessed over an ensemble of simulations and/or over a larger set of gridcells") is a separate thought and should have its own bullet.

**The first part of the bullet is not a result of our analysis, but a general statement. Whereas the second part that the reviewer mentions is the conclusion we draw. We have clarified the connection by reformulating the second sentence.**